# Acute Kidney Injury Biomarkers in Perioperative Care: A Scoping Review of Clinical Implementation

**DOI:** 10.3390/diagnostics16010094

**Published:** 2025-12-27

**Authors:** Konrad Zuzda, Paulina Walczak-Wieteska, Paweł Andruszkiewicz, Jolanta Małyszko

**Affiliations:** 1Department of Nephrology, Dialysis and Internal Medicine, Medical University of Warsaw, 02-901 Warsaw, Poland; jolanta.malyszko@wum.edu.pl; 2Doctoral School, Medical University of Warsaw, 02-091 Warsaw, Poland; 32nd Department of Anaesthesiology and Intensive Care, Medical University of Warsaw, 02-091 Warsaw, Poland; paulina.walczak-wieteska@wum.edu.pl (P.W.-W.); pawel.andruszkiewicz@wum.edu.pl (P.A.)

**Keywords:** AKI, biomarkers, PENK, CCL-14, TIMP-2*IGFBP-7, NGAL, perioperative care

## Abstract

**Background**: Acute kidney injury (AKI) remains one of the most common perioperative complications, carrying substantial mortality and healthcare burden. Traditional diagnostic criteria relying on serum creatinine and urine output are limited by delayed detection and inability to characterize the underlying injury phenotype. This scoping review examined the current state of novel AKI biomarker research in perioperative care, evaluated their clinical implementation, and identified knowledge gaps. **Methods**: A systematical search was performed for studies investigating novel AKI biomarkers in surgical settings. Biomarkers were categorized as functional, stress, or damage markers. Data extraction focused on diagnostic performance, clinical outcomes, regulatory approval status, and implementation barriers. A narrative synthesis was organized by biomarker category and thematic areas. **Results**: Several biomarkers demonstrated superior early diagnostic performance compared to traditional ones, including PENK or CCL-14, showing promising accuracy for AKI detection and outcome prediction. TIMP-2*IGFBP-7 and NGAL achieved regulatory approval, and biomarker-guided KDIGO care bundles significantly reduced AKI incidence in surgical populations. However, substantial heterogeneity exists in assays, cutoff values, and clinical validation across different clinical settings. **Conclusions**: Novel AKI biomarkers offer a promise for early detection and risk stratification in perioperative care, yet widespread clinical adoption requires addressing standardization challenges, establishing cost-effectiveness, and validating implementation strategies.

## 1. Introduction

Acute kidney injury (AKI) represents a syndrome characterized by rapid deterioration of kidney function. In perioperative care, it is associated with substantial short- and long-term complications, increased mortality [1], and high healthcare utilization, especially in persistent cases [2]. Current Kidney Disease Improving Global Outcomes (KDIGOs) diagnostic criteria (Table 1) for AKI rely on serum creatinine (SCr) and urine output (UOP) [3]. Debate continues regarding the limitations of conventional markers [4,5]. Specifically, they delay kidney injury recognition and show low sensitivity for early tubular injury, potentially missing cases where kidney damage occurs without reaching KDIGO diagnostic criteria [6]. This delayed detection precludes timely intervention during the critical window when preventive strategies are most effective, directly contributing to progression toward long-term poor kidney outcomes.

Over the past two decades, advances in biomarker research have identified novel markers that enable earlier AKI detection, better risk stratification, and improved understanding of AKI pathophysiology [7]. These biomarkers can be categorized based on their biological mechanisms: functional markers assessing glomerular filtration, stress markers reflecting cellular response to injury, and damage markers indicating structural injury [8]. Several biomarkers have achieved regulatory approval, including U.S. Food and Drug Administration (FDA) clearance, supporting their clinical translation [9,10,11]. Importantly, recent clinical trials have demonstrated that biomarker-guided implementation of KDIGO care bundles can significantly reduce AKI incidence and severity in high-risk surgical populations [12,13], establishing proof-of-concept for biomarker-driven precision medicine in perioperative care. Despite these advances, challenges remain in standardizing assays, determining optimal cutoff values, establishing cost-effectiveness, and integrating biomarkers into routine clinical workflows.

This scoping review examines the current state of AKI biomarker research, identifies critical knowledge gaps in clinical implementation, and provides recommendations for future research priorities to advance precision medicine approaches in perioperative AKI management.

## 2. Materials and Methods

Two researchers independently conducted a systematic search of EuropePMC, MEDLINE, and Scopus databases using controlled vocabulary and keywords related to AKI, novel biomarkers (NGAL, KIM-1, cystatin C, TIMP-2, IGFBP-7, L-FABP, IL-18, CXCL-9, CCL-14, hepcidin, PENK, DKK-3, alpha-1-microglobulin, and CHI3L-1), and surgery or perioperative care. Studies were included if they investigated novel AKI biomarkers in surgical patients, while critically ill populations, non-postprocedural AKI, and preclinical studies without clinical translation were excluded. Following duplicate removal, two independent reviewers screened titles and abstracts, followed by full-text review of potentially eligible studies. Disagreements were resolved through discussion or consultation with a third reviewer.

A narrative synthesis approach organized findings by biomarker category (functional, stress, damage) and thematic areas, including diagnostic performance, regulatory approval, and clinical adoption. Knowledge gaps were systematically identified by analyzing inconsistencies in findings, insufficient evidence areas, and discrepancies between preclinical data and clinical translation. Limitations include restriction to English-language publications, absence of formal quality assessment, possible underrepresentation of emerging biomarkers, and substantial methodological heterogeneity precluding quantitative synthesis. Additionally, non-standardized reporting of measurement timing and threshold values hindered direct comparisons, and publication bias may have overestimated biomarker diagnostic accuracy and clinical utility.

## 3. Current State of AKI Biomarker Research

The classical definition of AKI is primarily based on two key parameters, SCr and UOP. Both measures are well-established and cost-effective; however, they present notable limitations (Table 2) [14]. Moreover, current classification does not accurately differentiate the cause or site of renal injury. SCr is characterized by low sensitivity and specificity and is influenced by factors such as muscle mass, hepatic function, and fluid balance. Furthermore, elevations in SCr often occur 24–36 h after the onset of injury due to increased SCr half-life while eGFR decreases [15]. Similarly, UOP is subject to alterations from hypovolemia, fasting, or the administration of diuretics, which diminishes its reliability for evaluating true renal parenchymal injury during acute perioperative periods. Despite extensive clinical application, SCr has not yet been supplanted by newer biomarkers in the diagnosis of AKI [5]. To overcome the limitations of traditional AKI biomarkers during the perioperative period, extensive research is concentrated on identifying and applying new, early, and effective biomarkers that can better characterize the etiology and severity of renal injury [16] (Table 3). The inherent limitations of SCr in accurately reflecting glomerular filtration rate (GFR) underscore the need to refine the KDIGO definition and incorporate novel biomarkers into GFR estimation formulas. For example, a multicenter study demonstrated that the PENK-Crea equation provides superior accuracy in GFR estimation compared to most traditional and recently developed creatinine-based equations [17].

### 3.1. Functional Biomarkers and Surrogate Markers of the Glomerular Filtration Rate

#### 3.1.1. PENK

Proenkephalin A 119–159 (PENK), with a molecular weight of 4.5 kDa, is freely filtrated across the glomerular membrane, making its plasma concentration directly proportional to the GFR [18]. The precise role and function of enkephalins in the kidneys remain incompletely understood [19]. However, recent studies have suggested a potential regulatory role for enkephalins in the control of diuresis and natriuresis [20]. The importance of PENK in perioperative care has been confirmed by studies in cardiac surgery and contrast-induced (CI)-AKI settings. Elevated preoperative and postoperative PENK levels were associated with increased AKI risk. Changes in PENK serum levels predicted AKI development more rapidly than SCr [21,22,23] and hospital mortality after AKI [24]. However, there are discrepancies regarding whether PENK levels increase earlier than SCr levels [25]. Most studies have focused on high-risk vascular surgery procedures with small but homogeneous patient groups [19,25,26,27]. Meta-analytic evidence demonstrated that PENK achieves a pooled sensitivity of 0.69 and specificity of 0.76 for early AKI detection, with an optimal cutoff of approximately 57.3 pmol/L, with moderate sensitivity and specificity [19].

#### 3.1.2. IL-18

Interleukin-18 (IL-18) functions as a potent proinflammatory mediator in both innate and adaptive immune responses. In the context of AKI, IL-18 is produced primarily by proximal tubular epithelial cells following injury. The cytokine plays a direct pathogenic role in kidney injury by stimulating inflammatory pathways and promoting tubular cell apoptosis [28]. The absence of IL-18 has been shown to offer protection against tubular damage [29]. The usefulness of urinary IL-18 as a single-biomarker in clinical trials has not been confirmed despite its physiological basis. A prospective observational cohort study on IL-18 in cardiac surgery patients did not confirm the biomarker’s effectiveness as a predictor of AKI but instead linked changes in its concentration to the inflammatory process caused by the use of cardiopulmonary bypass (CBP) [30,31]. A comprehensive meta-analysis evaluating urinary IL-18 across multiple clinical settings found moderate predictive value, with pooled sensitivity of 0.64, specificity of 0.77, and an area under the receiver operating characteristic curve (AUC) of 0.78. In children and adolescents, compared with adults, AUC values were up to 0.78 and diagnostic odds ratios (DORs) were significantly higher than in adult populations [32]. A urinary IL-18 cutoff of 1477 pg/mg creatinine at 4 h postoperatively provided optimal discrimination for AKI in pediatric cardiac surgery [33]. In adults, no standardized numeric cutoff was established; AKI risk can assessed by relative increases or highest quintile values [34].

#### 3.1.3. L-FABP

Liver-type fatty acid-binding protein (L-FABP) is a small 14 kDa cytoplasmic protein expressed in renal proximal tubular epithelial cells and involved in fatty acid metabolism [35]. L-FABP is released in response to ischemic and oxidative stress, and its urinary levels rise within hours of renal insult [36]. In pediatric and adult cardiac surgery, urinary L-FABP levels increase as early as 4–6 h postoperatively in patients who develop AKI, with AUC values ranging from 0.72 to 0.81 for early AKI prediction, outperforming or complementing other biomarkers [36,37,38,39]. L-FABP elevation correlated with clinical severity, including longer CBP duration, higher postoperative SCr, and prolonged hospital stay [36,38]. In vascular surgery, L-FABP measured at 6 and 24 h postoperatively was associated with reduced UOP and impaired renal function. The cutoff value for urinary L-FABP in the perioperative AKI was typically in the range of 2226.5 μg/g creatinine at 0 h and 673.1 μg/g creatinine at 2 h postoperatively in adult cardiac surgery, with sensitivity and specificity around 80% for early AKI detection [40]. In pediatric cardiac surgery, a 24-fold increase in urinary L-FABP from baseline at 4 h postoperatively has been associated with AKI, yielding an AUC of 0.81, sensitivity of 71%, and specificity of 68% [37]. Meta-analyses confirmed that urinary L-FABP has moderate sensitivity and specificity for AKI diagnosis across perioperative settings, but performance varies by patient population and timing [41]. L-FABP is most useful as part of a biomarker panel for perioperative AKI risk stratification and early intervention, rather than as a standalone test due to lack of standardized assays, cutoff values, and the influence of nonrenal factors [41,42].

### 3.2. Stress Biomarkers

#### TIMP-2 and IGFBP-7

Tissue inhibitor of metalloprotease-2 (TIMP-2) and insulin-like growth factor-binding protein 7 (IGFBP-7) are two urinary cell cycle arrest markers commercially available. Levels of TIMP-2 and IGFBP-7 increase due to changes in tubular filtration, reduced reabsorption, and leakage. TIMP-2 is mainly secreted by distal tubule cells, while IGFBP-7 is primarily secreted by proximal tubule cells [43]. In high-risk surgical patients, including those undergoing major abdominal, vascular, and cardiac procedures, measurement of urinary TIMP-2*IGFBP-7 within hours after surgery can identify individuals at increased risk for developing moderate-to-severe AKI (stage 2–3) within the next 12–24 h. Diagnostic performance is robust, with AUC values ranging from 0.80 to 0.90 for moderate-to-severe AKI prediction [44,45,46,47]. A cutoff of 0.3 (ng/mL)^2^/1000 is commonly used for early risk identification, while higher values (>2) are associated with greater risk and need for Renal Replacement Therapy (RRT) [45]. However, their diagnostic efficacy still remains unclear, considering the impact of different underlying diseases, various surgical procedures, sampling times, and different cutoff values used for AKI diagnosis [43,48,49,50]. Moreover, a post hoc analysis of 337 patients from the PrevAKI single-center study revealed that SCr measured at the same time as TIMP-2*IGFBP-7 was superior in predicting AKI development regardless of severity. The AUC for this biomarker panel was 0.60, compared with 0.82 for SCr [51].

### 3.3. Injury and Damage Biomarkers

#### 3.3.1. C-C Motif Chemokine Ligand-14

Urinary C-C motif chemokine ligand-14 (CCL-14), also known as human C-C chemokines-1, is a small cytokine belonging to the chemokine family, primarily produced by macrophages and monocytes in response to renal injury. CCL-14 is released from injured renal tubular epithelial cells in response to inflammatory mediators and binds to C-C chemokine receptors on monocytes and T cells, driving differentiation into proinflammatory Th1 cells and activating proinflammatory macrophages that initiate downstream inflammatory cascades [52]. Elevated urinary CCL-14 perpetuates renal dysfunction through multiple mechanisms: promoting inflammation and fibrosis via macrophage-mediated pathways, inducing maladaptive repair responses that prevent functional restoration, and potentially triggering apoptosis of renal tubular cells through cell cycle modulation [52,53]. Persistently elevated urinary CCL-14 thus reflects an ongoing inflammatory phenotype characterized by defective renal repair mechanisms and increased risk of progression to prolonged AKI, CKD, and end-stage renal disease [54,55]. The landmark RUBY study [56] demonstrated that elevated urinary CCL-14 levels predict persistent and severe AKI (defined as KDIGO stage 3) with superior discriminative performance compared with other contemporary biomarkers. Subsequent validation studies and meta-analyses have consistently confirmed CCL-14’s clinical utility across diverse patient populations, establishing it as one of the most feasible biomarker for persistent AKI prediction [57,58]. A standardized clinical assay for urinary CCL-14 has established two clinically useful cutoff values. A CCL-14 concentration of 1.3 ng/mL achieved 91% sensitivity and identified the vast majority of patients who develop persistent severe AKI, with a negative predictive value of 92%, making it valuable for ruling out disease progression. Conversely, a cutoff of 13 ng/mL achieved 93% specificity and a positive predictive value of 72%, allowing for identification of patients at highest risk of deterioration [54]. Recent meta-analyses evaluating seven major AKI biomarkers across 31 studies found that CCL-14 demonstrated superior diagnostic efficacy with an overall AUC of 0.79, outperforming TIMP-2*IGFBP-7, sCysC (AUC 0.70), and NGAL (AUC 0.71). Notably, performance varied substantially by clinical context: in the postoperative population, CCL-14 demonstrated AUC of 0.83–0.93, substantially exceeding other biomarkers [57].

#### 3.3.2. C-X-C Motif Chemokine Ligand-9

C-X-C Motif Chemokine Ligand 9 (CXCL-9) is an interferon-γ (IFN-γ)-induced chemokine involved in lymphocyte chemotaxis with a size of ~14 kDa [59]. Because IFN-γ is crucial for recruiting activated T lymphocytes during interstitial inflammation, CXCL-9 might serve as a specific marker of tubulointerstitial inflammation [60]. CXCL-9 occupies a distinct diagnostic niche, as it was exclusively detected in conditions of ischemia–reperfusion injury, found in 29% of living donor kidney transplant recipients and 63% of donation after circulatory death recipients [61]. Notably, while other biomarkers increase in response to tubular stress and injury regardless of mechanism, CXCL-9 appears to be a marker of inflammatory activation, making it complementary rather than redundant to existing biomarkers [61].

CXCL-9 shows particular promise in acute interstitial nephritis (AIN), a specific form of AKI that has a more treatable etiology than tubular injury [62]. Urinary CXCL9 has demonstrated an AUC of up to 0.94 for AIN diagnosis. More specifically, CXCL-9 levels were 7.6-fold higher in patients with AIN compared with those with other forms of AKI, and notably, 8-fold higher when comparing AIN with acute tubular injury [60]. The combined use of CXCL-9 with tumor necrosis factor-α (TNF-α) and IL-9 demonstrated superior diagnostic performance compared to individual biomarkers alone in AIN detection [60].

#### 3.3.3. Dickkopf-Related Protein-3

Dickkopf-related protein 3 (DKK-3) is a 38 kDa secreted glycoprotein synthesized by stressed renal tubular epithelium. DKK-3 appears to inhibit the protective Wnt/β-catenin signaling pathway, which is transiently activated as a repair mechanism following tubular injury [63]. In a landmark study [64], preoperative urinary DKK-3 concentrations independently predicted the development of postoperative AKI. In this cohort of cardiac surgery patients, urinary DKK-3 concentrations relative to creatinine that exceeded 471 pg/mg were associated with significantly increased AKI risk. In other clinical settings, a prospective study of 490 patients undergoing coronary angiography found that subjects who developed CI-AKI had a 3.8-fold higher urinary DKK-3/creatinine ratio than those without CI-AKI (7.5 pg/mg vs. 2.0 pg/mg, *p* = 0.047). However, the diagnostic accuracy in this setting was more modest compared with cardiac surgery, with an AUC of 0.61. The best cutoff value for DKK3 was 1.7 pg/mg creatinine, achieving 47.4% sensitivity and 72.4% specificity [65].

#### 3.3.4. Chitinase 3-like Protein-1

Chitinase 3-like protein-1 (CHI3L-1) is a 39 kDa protein secreted by various cell types, including macrophages, epithelial cells, fibroblasts, and smooth muscle cells, with macrophages representing a key source of urinary CHI3L-1 during renal stress or injury. CHI3L-1 is considered a “repair phase” protein that becomes elevated in response to structural kidney injury and triggers renoprotective mechanisms through inhibition of apoptosis in renal epithelial cells and suppression of pyroptosis and inflammasome activation in macrophages [66].

In a prospective cohort study [67], which examined 203 cardiac surgery patients, it was found that urinary CHI3L-1 had inadequate predictive value for detecting AKI within 48 h after postoperative admission, with an inability to reliably distinguish patients who developed any stage of AKI. In contrast, SCr measurements obtained at 4 h after surgery in a mixed surgical and medical cohort showed superior performance with an AUC of 0.792 for predicting stage 1 or greater AKI, and when combined with baseline SCr changes, achieved an excellent AUC of 0.938 for predicting stage 2 or greater AKI within 12 h. CHI3L-1 demonstrated prognostic significance for AKI progression and mortality too; in a cohort of 249 AKI patients, urine CHI3L-1 concentrations ≥ 5 ng/mL were associated with disease progression or in-hospital mortality [68].

#### 3.3.5. Kidney Injury Molecule-1

Kidney Injury Molecule-1 (KIM-1) is a transmembrane glycoprotein with minimal expression in normal kidney tissue but is more abundant in injured proximal tubules. Proteolytic cleavage releases its extracellular domain into urine. KIM-1 has been demonstrated to plays a role in both the process of kidney injury, mostly of ischemic or nephrotoxic drug origin, and the subsequent recovery processes [69]. A systematic review of patient data from various settings, including perioperative ones, reported urinary KIM-1 diagnostic sensitivity and specificity for AKI as 74% and 84%, respectively [70]. In an AKI biomarker prospective study investigating the utility of urinary biomarkers of AKI in major abdominal surgery, the AUC of KIM-1 was statistically significant for stage 1 AKI 0.68, but not for stage 2 [44]. In pediatric cohorts, KIM-1 showed moderate utility in forecasting the need for RRT in pediatric AKI, with an AUC of 0.71 [71]. According to a meta-analysis, a urinary concentration cutoff of 19 ng/for CI-AKI following cardiac catheterization yielded an AUC of 0.88, with sensitivity of 84% and specificity of 78% for predicting AKI [72]. Elevated preoperative KIM-1 levels predicted long-term adverse events, including death, cardiovascular events, and chronic kidney disease (CKD) progression, in cardiac surgery patients [73].

#### 3.3.6. Cystatin C

Cystatin C (CysC) is a 13 kDa, 122 amino acid cysteine protease inhibitor. The molecule is freely filtered at the glomerulus, completely reabsorbed by the proximal tubule, and undergoes full catabolism intracellularly with no return to the bloodstream and no active tubular secretion. CysC concentration is independent of age, gender, muscle mass, and nutritional status, with a relatively constant production rate across populations [74]. Serum CysC (sCysC) serves as a marker of glomerular filtration, while elevated urinary CysC may indicate proximal tubular dysfunction independently of changes in filtration rate [75]. In pediatric cardiac surgery, sCysC peaks early at approximately 8 h postoperatively [76]. Multiple studies demonstrated that sCysC measured within 6 to 24 h postoperatively provides robust predictive capability for subsequent AKI development [77,78,79]. Systematic review and meta-analysis demonstrated that sCysC exhibits high diagnostic accuracy for postcardiac surgery AKI, with sensitivity of 0.67, specificity of 0.87, and an AUC of 0.86 [79]. A network meta-analysis comparing multiple biomarkers ranked sCysC among the highest performers, with a hierarchical summary receiver operating characteristic value of 0.82 [80]. In contrast, a multicenter study focusing on children [67] utilized urinary CysC to define AKI substages, finding that subclinical AKI (sAKI); CysC-positive without KDIGO AKI occurred in 20.2% of non-AKI patients and was associated with a mortality risk close to that of CysC-negative AKI substage A. Ultimately, this subphenotyping confirmed that CysC-positive AKI substage B patients, representing 50% of traditional AKI cases, were more likely to develop severe AKI (stage 3) and were associated with the highest 30-day risk. In pediatric cardiac surgery context, reported sCysC cutoff for AKI was >1.33 mg/L at 6 h postoperatively [78]. A postoperative increase ≥ 25% from baseline also served as a threshold for AKI detection in adults [81]. Despite advantages over SCr for AKI detection, CysC has notable limitations. Glucocorticoids may increase CysC production independently of renal function [82]. Thyroid dysfunction [83,84], diabetes, high C-reactive protein and white blood cell counts, and low serum albumin were also associated with higher levels of CysC [85]. This can lead to clinically relevant discrepancies of up to 15.3% difference in mean eGFR [86].

#### 3.3.7. Neutrophil Gelatinase-Associated Lipocalin

Neutrophil gelatinase-associated lipocalin (NGAL) is a protein expressed by various cells, including neutrophils and renal tubular cells. It exists in monomeric, homodimeric, and heterodimeric forms. NGAL plays a role in iron metabolism and exhibits antimicrobial properties. NGAL concentrations rise in response to epithelial injury and inflammation [87]. In the context of AKI, NGAL levels in blood and urine increase within 2–6 h following kidney injury and peak between 6 and 12 h, significantly earlier than the rise in SCr [88]. The diagnostic performance of NGAL in the postoperative period has been evaluated in several meta-analyses with various surgical procedures, including both noncardiac and cardiac surgery [89,90,91]. In a study of cardiac surgery patients with postoperative NGAL measurement, analysis demonstrated an AUC of 0.71 with a cutoff of 154 ng/mL, achieving sensitivity of 76% and specificity of 59% [92]. In another study with patients undergoing CPB, NGAL levels greater than 353.5 ng/mL were independently associated with postoperative AKI [93]. In pediatric cardiac surgery patients, plasma NGAL at 2 h after CPB demonstrated discriminative performance with an AUC of 0.96, sensitivity of 0.84, and specificity of 0.94 using a cutoff value of 150 ng/mL [94]. Urine NGAL has been demonstrated to serve as an independent predictor of the development of AKI and the subsequent necessity for RRT in liver transplant patients [91] and mortality after coronary interventions [95]. Nevertheless, the clinical utility of the biomarker is supported by only weak evidence [96].

#### 3.3.8. Alpha-1-Microglobulin

Alpha-1-microglobulin (α1m) represents a marker of proximal tubular damage in the context of acute kidney injury. A low-molecular-weight protein synthesized in liver is freely filtered by the glomerulus and normally reabsorbed and metabolized by the proximal tubule [97]. In the TRIBE-AKI study of 1464 adults undergoing cardiac surgery [98], preoperative urinary α1m was independently associated with postoperative AKI, with each twofold higher preoperative concentration conferring an adjusted odds ratio of 1.36 for AKI development. Furthermore, preoperative α1m successfully identified patients at elevated risk for adverse long-term outcomes, including CKD incidence and progression, cardiovascular events, and all-cause mortality during a median 6.7-year follow-up period [98]. While baseline urinary α1m showed predictive value, postoperative measurements on day 1 did not reliably detect in-hospital AKI [98].

α1m’s protective properties derive from its reactive cysteine residue (C34), enabling antioxidant, heme-binding, and radical-scavenging activities. These mechanisms function as “tissue cleaning” processes during oxidative stress and ischemia–reperfusion injury, and α1M further maintains mitochondrial energy balance during cellular injury [99]. RMC-035, a recombinant α1m, has demonstrated early signals of renal protection in a Phase 1 trial [100]. Although a Phase 2 study did not reduce AKI at 72 h, it showed a significant decline in major adverse kidney events (MAKEs) at 90 days, driven primarily by reduced persistent renal dysfunction, warranting further evaluation in ongoing Phase 2b studies [101].

#### 3.3.9. Hepcidin-25

Hepcidin-25 is a low-molecular-weight peptide (2.78 kDa) that is freely filtered by the glomerulus, with approximately 97% reabsorbed by the proximal tubule under physiological conditions. This iron-regulatory peptide hormone possesses antimicrobial properties and is synthesized primarily in the liver, with additional expression in renal tissue [102]. Cardiac surgery with CPB induces hemolysis and tissue injury, releasing free hemoglobin and myoglobin that increase circulating labile iron, which promotes ferroptosis, an iron-dependent form of cell death that damages renal tubular epithelial cells [103]. By sequestering iron intracellularly, hepcidin-25 may limit oxidative stress and ferroptotic injury, providing a renoprotective mechanism in ischemia–reperfusion injury [104].

Early postoperative urinary hepcidin demonstrates significant diagnostic value for predicting AKI protection following coronary interventions [102] and cardiac surgery. In a 100-patient study, urinary hepcidin-25 levels were three to seven times higher in AKI-free patients at 6 and 24 h post-CPB (AUC 0.80–0.88), with creatinine-adjusted hepcidin-25 at 6 h independently predicting AKI avoidance [105]. In a larger prospective cohort of 306 patients, elevated urinary hepcidin-25 on postoperative day 1 inversely predicted AKI development and, when combined with baseline GFR and diabetes status, improved overall AKI prediction (AUC 0.82), outperforming conventional clinical scores [103]. The plasma NGAL/hepcidin-25 ratio has been as a promising marker for predicting MAKE in cardiac surgery patients [106].

#### 3.3.10. Gamma-Glutamyltransferase

Gamma-glutamyltransferase (GGT), also known as gamma-glutamyl transpeptidase, is a membrane-bound enzyme predominantly expressed on proximal tubular epithelial cells of the kidney. In contrast-induced AKI in patients undergoing coronary procedures, GGT showed elevated levels in patients who developed AKI compared to controls, with an odds ratio of 3.21 (95% CI: 1.26–8.15). However, the evidence base demonstrated significant heterogeneity across studies, suggesting variability in GGT’s predictive performance depending on clinical context and patient population [107].

#### 3.3.11. π-Glutathione S-Transferase

π-Glutathione S-transferase (π-GST) is a preformed cytoplasmic enzyme that serves as a site-specific marker of distal renal tubular injury. As a constitutive detoxification enzyme, π-GST comprises approximately 2% of soluble protein in renal tubules and is exclusively released into urine when cellular integrity of the distal tubule is compromised. In a multicenter prospective study of 141 cardiovascular surgical patients [108], urinary π-GST demonstrated superior predictive for identifying advanced AKI (stage 2 or 3). π-GST measured at 3 h post-surgery achieved an AUC of 0.784, with an optimal cutoff value of 16.5 μg/L providing 75.0% sensitivity and 68.2% specificity. For predicting the composite outcome of advanced AKI or in-hospital mortality, the optimal cutoff was 14.7 μg/L at 3 h post-surgery, yielding 73.3% sensitivity and 66.7% specificity. Additional studies have confirmed that π-GST performs best during the early postoperative period from 3 to 12 h [109].

#### 3.3.12. Interleukin-9

Interleukin-9 (IL-9) is a cytokine, traditionally associated with allergic and type 2 immune responses. Urine IL-9 is the most extensively characterized biomarker for diagnosing AIN, a form of AKI typically requiring kidney biopsy. In a cohort of 218 patients undergoing biopsy, urine IL-9 demonstrated superior diagnostic accuracy (AUC 0.84) compared to clinical assessment (AUC 0.62), with a high-specificity cutoff yielding a 0.94 post-test probability for AIN and the potential to reduce unnecessary biopsies [110]. IL-9 produced by a subset of CD4+ T helper cells promotes mast cell infiltration and degranulation in AIN, triggering TNF-α release and kidney inflammation, as evidenced by increased TNF-α-positive cells and mast cells in AIN biopsies [110]. In one small observational study examining plasma NGAL and IL-9 as predictors of postoperative AKI in 21 coronary artery bypass patients undergoing CPB, neither biomarker achieved statistical significance as an independent predictor, suggesting limited usefulness of IL-9 in other clinical context [111].

#### 3.3.13. Monocyte Chemoattractant Peptide-1

Monocyte chemoattractant protein-1 (MCP-1) is an inflammatory biomarker that serves to predict AKI and long-term kidney outcomes. MCP-1 belongs to the chemokine family and functions as a pivotal mediator of both innate immune responses and tissue inflammation. Stimulation of renal tissue results in a pronounced upregulation of MCP-1 expression, which corelates strongly with the severity of kidney injury [112]. Through the activation of monocytes, MCP-1 contributes to the progression of inflammation and may ultimately lead to renal failure. Moreover, inflammatory cytokines can increase MCP-1 secretion from renal tubular epithelial cells. Follow-up studies on post-AKI patients have demonstrated that elevated MCP-1 is significantly associated with an increased risk of developing CKD [113]. In the TRIBE-AKI study on cardiac surgery patients, elevated urinary MCP-1 was related to an increased incidence of AKI, later development of CKD, and higher mortality. MCP-1 independently demonstrated utility as a reliable biomarker of tubular injury [114,115].

#### 3.3.14. Netrin-1

Netrin-1 (NTN-1) is a large 72 kDa antiinflammatory protein predominantly secreted by proximal tubule epithelial cells in response to hypoxic or toxic injury. NTN-1 contributes to vascular patterning and maturation during renal development, with the kidney exhibiting some of the highest levels of NTN-1 expression. Notably, in animal models of ischemic AKI, NTN-1 was detectable in urine at an early stage following injury [116]. A few human studies involving pediatric patients after cardiac surgery described discrepancies in the effectiveness of NTN-1 as an early biomarker of AKI [116,117]. A 6 h post-CPB urinary NTN-1 concentration of approximately 2462 ± 370 pg/mg creatinine was observed in AKI cases, compared to lower values in controls, but no specific diagnostic cutoff was defined [117]. In liver transplantation, a 2 h postoperative urinary NTN-1 value of 897.8 ± 112.4 pg/mg creatinine was associated with AKI [118].

#### 3.3.15. Semaphorin-3A

Semaphorin-3A (SEMA-3A) is a 65 kDa protein that plays roles in the regulation of angiogenesis, organogenesis, and immune responses. It has been identified in adult podocytes and collecting tubules. While undetectable in the urine of healthy individuals, SEMA-3A can be measured within hours following ischemia–reperfusion injury, secreted by injured podocytes and distal tubular cells. Inactivation of SEMA-3A suppresses this secretion [119]. In cases of ischemia–reperfusion AKI, SEMA3A mediates tissue damage by promoting inflammation and apoptosis of tubular epithelial cells. SEMA-3A has been identified as a potential biomarker for the prediction of contrast-induced AKI [120]. A study in patients undergoing percutaneous coronary intervention (PCI) reported SEMA-3A cutoff of 389.5 pg/mg creatinine at 2 h post-insult, with 94% sensitivity and 75% specificity [120]. In liver transplantation, 2 h postoperative mean values of 847.9 ± 93.3 pg/mg creatinine were associated with AKI development [118]. Pediatric cardiac surgery patients demonstrated peak values of 2596 ± 591 pg/mg creatinine at 6 h post-CPB, although a specific diagnostic threshold was not established [121].

#### 3.3.16. Osteopontin

Osteopontin (OPN) is phosphorylated 37.7 kDa glycoprotein and is predominantly synthesized in kidney tissue. In the context of ischemia–reperfusion injury, OPN plays a complex dual role as both a pathogenic and protective mediator [122]. One study suggested that increased OPN release following AKI is associated with neurocognitive decline [123]. Furthermore, circulating OPN released from the kidneys during AKI has been implicated in the development of remote lung inflammation and subsequent respiratory failure [124]. OPN was also associated with postoperative complications and inflammatory responses after major surgery [125]. In the CASABLANCA AKI Prediction substudy, OPN was evaluated alongside KIM-1, IL-18, and CycC in patients undergoing coronary and peripheral angiography. Incorporating OPN into traditional risk scoring models enhanced prediction of both procedural AKI and long-term cardiorenal outcomes during a median 3.7-year follow-up [126].

### 3.4. Renal Tubular Epithelial Cells

Oyaert et al. [127] prospectively evaluated automated urinary flow cytometry measurement of renal tubular epithelial cells (RTECs) as an early AKI biomarker in 239 adult cardiac surgery patients. The study found that RTEC counts at 12 and 24 h post-admission demonstrated excellent diagnostic accuracy (AUC 0.946 at 24 h for all AKI up to 7 days), performing comparably to or superior to TIMP-2*IGFBP-7, GGT, and α1m. Notably, RTEC counts correlated with AKI severity, distinguished between rapid reversal and persistent AKI, and showed significantly higher levels in patients requiring RRT [128].

### 3.5. Factors Affecting Biomarker Specificity: False Positives and False Negatives

Clinical interpretation of novel AKI biomarkers requires awareness of conditions that may confound their diagnostic accuracy. NGAL concentrations are elevated in systemic infections and inflammatory states independently of kidney injury [87], while urinary IL-18 rises in response to CPB-induced inflammation rather than tubular injury specifically [30,31]. CysC levels are affected by glucocorticoid therapy, thyroid dysfunction, diabetes, and hypoalbuminemia [82,83,84,85]. TIMP-2 and IGFBP-7 may be elevated in any condition causing cellular stress [43,48,49,50]. False-negative results may occur when sampling times do not align with biomarker kinetics [129]. These limitations necessitate clinical context-based biomarker interpretation and support adoption of multi-biomarker panels.

## 4. Biomarker Panel

A multi-biomarker panel for AKI represents a promising approach to enhance the detection of kidney impairment by integrating multiple markers and thereby overcoming the limitations of single-biomarker strategy, including insufficient sensitivity for early AKI detection and inability to distinguish AKI pathophysiology [130]. Different biomarker signatures can characterize distinct AKI subphenotypes with prognostic implications. For instance, combinations of NGAL and IL-18 effectively differentiated acute tubular nephritis from prerenal AKI [131], CXCL-9 with TNF-α and IL-9 demonstrated superior diagnostic performance in AIN diagnosis [60], while CysC combined with NGAL provided predictive information regarding disease progression risk [132]. The FDA-approved TIMP-2*IGFBP-7 panel represents the most clinically implemented combined biomarker strategy, particularly in perioperative and critical care settings. Expert consensus indicated optimal application in patients undergoing major surgery or those with hemodynamic instability [133]. Additionally, a multiplex panel with 21 serum and urinary proteins, including biomarker candidates, demonstrated enhanced discrimination for AKI versus CKD phenotypes and enabled stratification of progression risk [134].

## 5. Regulatory Approval and Clinical Adaptation

### Regulatory Approval Status

Approval of AKI biomarkers for commercial use is progressing slowly. To date, only a few patents have been granted for clinical use by European or American drug regulatory authorities. KIM-1, OPN, NGAL, and CysC have been approved by the FDA [135] and KIM-1 and CysC by the European Medicines Agency (EMA) for the detection of drug-induced kidney injury during nonclinical and clinical drug trials [136].

The FDA has formally endorsed the utilization of NephroCheck^®^ (TIMP-2*IGFBP-7) as a biomarker for the purpose of predicting the risk of AKI, particularly within critical care settings [137]. ProNephro AKI (NGAL), an immunoassay for the quantitative determination of NGAL in urine that recently received FDA clearance as well [87]. The penKid (PENK) assay has not yet received full approval; however, it has obtained CE-IVD certification for specific diagnostic platforms [138].

A key advancement in AKI diagnostics would be the integration of novel biomarkers into point-of-care testing panels, enabling rapid and bedside evaluation of renal function for early detection and management of AKI. Still, the expense associated with quantifying the concentration of a single-biomarker or biomarker pair considerably exceeds that of conventional biomarker assessments.

## 6. AKI Care Bundle Implementation and Biomarker Panels

The KDIGO guidelines [3], established in 2012, outline a series of recommendations aimed at reducing the occurrence of AKI. These guidelines emphasize avoiding nephrotoxic agents and discontinuing angiotensin-converting enzyme inhibitors and angiotensin receptor blockers for 48 h before surgical procedures. Moreover, they highlight the importance of closely monitoring classic AKI biomarkers, preventing hypo- and hyperglycemia and radiocontrast-induced nephropathy, and continuously tracking and optimizing hemodynamics. They also advocate for optimization of patient’ volume status. These recommendations were mainly based on expert consensus rather than multicenter trial results, and they were developed at a time when new AKI biomarkers were just being introduced into research practice [3]. Since then, several meta-analyses have shown that implementing AKI care bundles in hospitalized patients during routine clinical practice can effectively improve outcomes for those diagnosed with AKI or at risk of developing it [13]. Similar findings were observed in cardiac surgery patients [12]. A meta-analysis of 16,540 patients demonstrated that care bundles incorporating biomarkers had greater impact than bundles without biomarkers, reducing the risk of MAKE, AKI, and the need for RRT, although only urinary TIMP-2·IGFBP-7 and NGAL were evaluated [139]. Biomarker selection for this meta-analysis was likely driven by regulatory clearance and availability of substantial clinical trial data. However, subsequent meta-analysis examining 16 studies with 25,690 patients indicated more modest effects, with AKI incidence reduced only in studies utilizing novel biomarkers, electronic alerts, or risk prediction scores (OR 0.71; 95% CI 0.53–0.96) [13].

PrevAKI [140,141] and BigpAK [142] trials have demonstrated reductions in AKI rates. TIMP-2*IGFBP-7 panel was investigated in the international, randomized controlled, multicenter BigpAK-2 trial [143]. This study tested a biomarker-guided approach with implementation of the KDIGO bundle after major surgery in 1180 patients across 34 European hospitals. Patients in the intervention group received KDIGO-recommended nephroprotective care, while the control group received usual care. The trial demonstrated that moderate or severe AKI occurred in 14.4% of intervention patients versus 22.3% of controls (OR 0.57, 95% CI 0.40–0.79; *p* = 0.0002), with a number needed to treat of 12 (95% CI 7–33). This decisive study confirmed that applying the KDIGO bundle to high-risk patients identified by biomarkers can significantly lower AKI incidence after major surgery without increasing adverse events [144].

Integrating novel biomarkers into AKI bundles may enhance diagnosis (Figure 1), particularly of sAKI, where elevated biomarkers precede traditional criteria yet indicate progression risk. Biomarker-guided implementation of KDIGO guidelines [145] has allowed earlier, targeted interventions. However, adding biomarkers to care bundles has not significantly reduced RRT requirements or mortality [139]. Recent studies using cluster analysis on comprehensive biomarker panels have identified distinct AKI subphenotypes with differing clinical characteristics, treatment responses, and outcomes. Four distinct AKI subphenotypes were identified, with subphenotypes 3 and 4 showing markedly different prognoses and treatment requirements [146,147].

**Figure 1 diagnostics-16-00094-f001:**
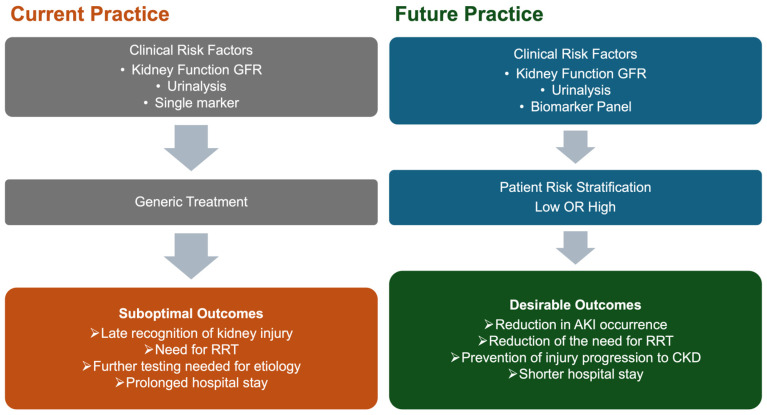
Current and future practice in managing acute kidney injury in perioperative settings. Abbreviations: GFR—glomerular filtration rate, RRT—renal replacement therapy, AKI—acute kidney injury, and CKD—chronic kidney disease.

A modeling study of hospitalized UK adults at risk of AKI assessed the diagnostic accuracy and cost-effectiveness of NephroCheck and NGAL biomarker tests used alongside standard care for AKI detection, compared with standard care alone. The evidence was insufficient to support the cost-effectiveness of widespread biomarker testing [148]. A systematic review and cost-effectiveness analysis by the UK’s National Institute for Health and Care Excellence (NICE) examining biomarkers for AKI risk assessment in critically ill conditions reached similar conclusions [149].

## 7. Conclusions

The expanding repertoire of AKI biomarkers offers opportunities for early detection, risk stratification, and physiological understanding of perioperative kidney injury, with regulatory-approved markers NGAL and TIMP-2*IGFBP-7, along with emerging candidates like CCL-14 or PENK demonstrating clinical utility when integrated into structured care bundles. However, translating biomarker discoveries into routine clinical practice remains challenging due to assay standardization issues, uncertain cost-effectiveness, and the complexity of integrating biomarker panels into time-sensitive perioperative decision-making.

## Figures and Tables

**Table 1 diagnostics-16-00094-t001:** The KDIGO (Kidney Disease: Improving Global Outcomes) criteria and staging of acute kidney injury.

AKI Stage	Serum Creatinine	Urine Output
1	1.5–1.9 times baseline (7 days) OR ≥0.3 mg/dL increase within 48 h	<0.5 mL/kg/h for 6–12 h
2	2.0–2.9 times baseline	<0.5 mL/kg/hour for ≥12 h
3	3.0 times baseline OR increase to ≥4.0 mg/dL OR initiation of RRT ^1^ OR decrease in eGFR ^2^ to <35 mL/min/1.73 m^2^ in patients < 18 years	<0.3 mL/kg/hour for ≥24 h OR anuric for ≥12 h

^1^ RRT: renal replacement therapy; ^2^ eGFR: estimated glomerular filtration rate.

**Table 2 diagnostics-16-00094-t002:** Key limitations of traditional markers of acute kidney injury.

Parameter	Limitations
Serum Creatinine	Low sensitivity for early AKI, requires ~50% loss of renal function before detectable elevation. Delayed rise 24–48 h after acute insult. Poor discrimination between acute and chronic changes. Volume status affects hemoconcentration. Pregnancy-related decrease during gestation. Muscle mass dependent varies with age, sex, body composition, and nutritional status. Influenced by nonrenal factors including dietary creatine, hepatic function, and metabolic rate. Medication interference with tubular secretion inhibitors falsely elevates levels. Non-specific for injury type including glomerular versus tubular. Inadequate for real-time monitoring. Unreliable in extreme body compositions.
Urine Output	Transient oliguria occurs with dehydration, stress, or volume depletion. Prone to collection errors even with urinary catheters (incomplete collection, spillage, miscalculation). Reduced output may reflect prerenal factors rather than intrinsic renal damage. Requires accurate hourly measurement and weight-based calculations. Normal or high urine output can occur despite significant renal dysfunction in non-oliguric injury. Requires sustained reduction over hours to meet diagnostic criteria. Influenced by fluid balance, medications, and hemodynamic status.

**Table 3 diagnostics-16-00094-t003:** Summary of AKI biomarkers’ characteristics in perioperative care.

Biomarker	Name	Sample	Detection Window	Cutoff Value
PENK	Proenkephalin A 119–159	Serum	2–6 h	~57.3 pmol/L
IL-18	Interleukin-18	Urine	4 h	1477 pg/mg creatinine
L-FABP	Liver-Type Fatty Acid–Binding Protein	Urine	4–6 h	~673.1 μg/g creatinine
TIMP-2*IGFBP-7	Tissue Inhibitor of Metalloproteinase-2 * Insulin-Like Growth Factor-Binding Protein-7	Urine	4–12 h	>0.3 (ng/mL)^2^/1000
CCL-14	C-C Motif Chemokine Ligand-14	Urine	6–24 h	1.3 ng/mL
CXCL-9	C-X-C Motif Chemokine Ligand-9	Urine	Postprocedural	N/S
DKK-3	Dickkopf-related Protein-3	Urine	Preprocedural	>471 pg/mg creatinine
CHI3L-1	Chitinase 3-like Protein-1	Urine	Postprocedural	≥5 ng/mL
KIM-1	Kidney Injury Molecule-1	Urine	12–24 h	19 ng/mL
CysC	Cystatin C	Urine, Serum	6–24 h	>1.33 mg/L (peds) ≥25% increase (adults)
NGAL	Neutrophil Gelatinase-Associated Lipocalin	Urine, Serum	2–6 h	~150–154 ng/mL
α1m	Alpha-1-Microglobulin	Urine	Preprocedural	N/S
Hepcidin-25	Hepcidin-25	Urine, Serum	6–24 h	N/S
GGT	Gamma-Glutamyltransferase	Urine	Postprocedural	N/S
π-GST	π-Glutathione S-Transferase	Urine	3–12 h	16.5 μg/L
IL-9	Interleukin-9	Urine	Postprocedural	N/S
MCP-1	Monocyte Chemoattractant Protein-1	Urine	Post-AKI	N/S
NTN-1	Netrin-1	Urine	2–6 h	~898–2462 pg/mg creatinine
SEMA-3A	Semaphorin-3A	Urine	2–6 h	~390–848 pg/mg creatinine
OPN	Osteopontin	Serum	Postprocedural	N/S
RTECs	Renal Tubular Epithelial Cells	Urine	12–24 h	N/S

Abbreviation: N/S—non-specified.

## Data Availability

No new data were created or analyzed in this study.

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
