# Peer review of "Acute Kidney Injury Biomarkers in Perioperative Care: A Scoping Review of Clinical Implementation"

_diagnostics, 2025, doi:10.3390/diagnostics16010094_

Round 1

Reviewer 1 Report

Comments and Suggestions for Authors

Dear authors, Many thanks for your interesting manuscript. The topic is a common clinical scenario and is very interesting indeed. I have highlighted several points in the uploaded pdf. 

Moreover, the basic criticism I have for your paper is that it needs some degree of restructuring. You have many biomarkers (maybe a fewer bit with more in depth explaining would make the paper an easier read), and I would suggest you structure them from the most studied to the least studied/more novel. An expansion is also probably needed in your final section towards future directions

there is a lack of addressing confounders - eg. Hepcidin is an iron metabolism protein and acutely post op iron metabolism changes / how is that reflected? I have noted it a few times on the manuscript

Author Response

We would like to thank the Reviewer for the opportunity to revise our manuscript. We are grateful for the insightful comments and constructive criticism. We have carefully considered each point and have revised the manuscript accordingly to improve its clarity, depth, and clinical relevance.

Below, we've included a point-by-point response to the Reviewer's comments.

Reviewer:
"The basic criticism I have for your paper is that it needs some degree of restructuring. You have many biomarkers (maybe a fewer bit with more in depth explaining would make the paper an easier read), and I would suggest you structure them from the most studied to the least studied/more novel. An expansion is also probably needed in your final section towards future directions there is a lack of addressing confounders - eg. Hepcidin is an iron metabolism protein and acutely post op iron metabolism changes / how is that reflected? I have noted it a few times on the manuscript."
Response:
We sincerely appreciate the reviewer’s suggestion regarding the manuscript’s structure. We gave significant thought to reordering the biomarkers from "most studied" to "least studied." However, after careful consideration, we respectfully believe that retaining the current structure, based on the biological classification framework (as cited in reference 8, Ostermann et al.), is most appropriate for this review.
Our rationale is that a mechanistic classification helps clinicians understand the pathophysiology of AKI more effectively:
Functional Biomarkers (e.g., PENK, IL-18, L-FABP) assess glomerular filtration.
Stress Biomarkers (e.g., TIMP-2*IGFBP-7) reflect the cellular response to injury.
Injury and Damage Biomarkers (e.g., NGAL, KIM-1, Hepcidin, etc.) indicate structural injury.

Regarding the confounders (including Hepcidin and iron metabolism), we agree that this is a critical area. We've significantly expanded our discussion of non-renal conditions and confounders in the revised manuscript. We have also revised Section 6. on future directions as suggested.

Reviewer:
"It is indeed the very definition of subclinical that relies on the existence of these biomarkers so it should not be used as a argument it is an aftermatch."
Response:
We agree with the reviewer that the phrasing regarding "subclinical AKI" needed refinement to avoid circular reasoning. We have revised the text in the Introduction to focus on the specific limitations of conventional markers regarding sensitivity and timing.

Reviewer:
"A flowchart could help - how many papers, identified, rejected etc."
Response:
We thank the reviewer for this suggestion. However, we respectfully believe a PRISMA-style flowchart is not appropriate for our manuscript. Our methodology did not follow strict systematic review protocols. Presenting a PRISMA-style diagram might imply a level of methodological rigor that narrative reviews are not designed to achieve, potentially misleading the reader. We have ensured that our search strategy and literature selection approach are transparently described in the text of the Materials and Methods section.

Reviewer:
"lacks of insight on how these markers might be raised or go down in other diseases (FP/FN)."
Response:
We agree that discussing false positives (FP) and false negatives (FN) caused by non-renal conditions is essential for clinical interpretation. We have added a discussion addressing these confounders at the end of Section 3., specifically highlighting NGAL, IL-18, CysC, and TIMP-2*IGFBP-7.

Reviewer:
"When do you say early, how much sooner & is it preventable? Any studies showing NNT?"
Response:
Thank you for this valuable comment. We have updated the manuscript to include specific data on the "Number Needed to Treat" (NNT) derived from recent major trials in Section 6.

Reviewer:
"NGAL / probably the best studied biomarker - could be discussed first and potentially to a larger extent."
"Cystatin-C / not a very novel marker, has been around for some time - could definitely expand."
"Alpha-1-microglobulin / could also go higher up."
Response:
We have maintained the mechanistic classification to preserve the biological narrative of the review. Cystatin C, while established, functions primarily as a marker of glomerular filtration though we acknowledge its tubular handling. We believe that grouping these by mechanism helps the reader understand why they are used, rather than simply how long they have been studied.

Reviewer:
"What about the metabolic disturbances to iron metabolism post op? Ferritin is an acute phase protein."
Response:
We appreciate the Reviewer highlighting the distinction between iron metabolism markers. We have clarified the text to distinguish Hepcidin's role from Ferritin, particularly regarding the confounding effect of inflammation.

We hope that the revisions and detailed responses provided here satisfactorily address the Reviewer's concerns. We thank the reviewer again for their time and valuable feedback, which has significantly strengthened our manuscript. We look forward to your positive response.

Reviewer 2 Report

Comments and Suggestions for Authors

AKI, or acute kidney injury, is a sudden and often reversible loss of kidney function that happens over hours or days. It is characterized by the kidneys' inability to filter waste from the blood, leading to a buildup of waste products and fluid in the body. AKI is frequently a complication of another serious illness and is defined by criteria such as a significant increase in serum creatinine or a sharp drop in urine output. However, standard biomarkers such as creatinine and BUN often indicate kidney problems late. Today, there is a need to investigate and introduce early biomarkers for the detection of AKI into clinical practice. This is exactly what the authors of this article have tried to inform us about.
This review paper represents a good overview and cross-section of the current situation. However, before accepting the paper, I would like to make a few suggestions.

Introduction section-  line 43 “Over the past two decades, extensive research has identified new biomarkers that enable earlier AKI detection, better risk stratification, and improved understanding of AKI pathophysiology.” -References should be added here, including references describing animal studies that contributed to the discovery of early biomarkers of AKI.

The introductory part itself should be expanded and further clarified the importance of incorporating new biomarkers into clinical practice.

Materials and methods are well written. The study is clear; the number of databases satisfactory for this type of study, as well as the number of biomarkers included in the study.

The description of the individual markers is satisfactory; however, I am missing an explanation of the introduction of the panel into clinical practice. The answer to the question “Why is it important?”

Table 2 is not clear and needs to be fixed. The way the table is written confuses the reader and does not clearly divide the columns and rows. It is suggested that the "center" format should not be used, but the Justify format.

The text should emphasize the importance of harmonizing and coordinating The KDIGO guidelines with clinical practice.

The conclusion follows the development of the text, and the number of references indicates the comprehensive work of the author. The preparation of the text was careful and thorough. However, I would suggest that the authors do another little research. Introducing these markers into the standard procedure would cost. is there a possibility of a "cost and benefit" analysis of the introduction of new biomarkers. I certainly believe that everything that can save a life should enter clinical practice, but unfortunately, we live in a capitalist world, so the data on the cost of introducing these protocols is very important.

The work should be accepted after minor corrections.

Author Response

Reviewer:
AKI, or acute kidney injury, is a sudden and often reversible loss of kidney function that happens over hours or days. It is characterized by the kidneys' inability to filter waste from the blood, leading to a buildup of waste products and fluid in the body. AKI is frequently a complication of another serious illness and is defined by criteria such as a significant increase in serum creatinine or a sharp drop in urine output. However, standard biomarkers such as creatinine and BUN often indicate kidney problems late. Today, there is a need to investigate and introduce early biomarkers for the detection of AKI into clinical practice. This is exactly what the authors of this article have tried to inform us about.
This review paper represents a good overview and cross-section of the current situation. However, before accepting the paper, I would like to make a few suggestions.

Response:
We thank the Reviewer for the positive assessment and appreciation of our manuscript. We have addressed the specific suggestions below.

Reviewer:
"Introduction section-  line 43 “Over the past two decades, extensive research has identified new biomarkers that enable earlier AKI detection, better risk stratification, and improved understanding of AKI pathophysiology.” -References should be added here, including references describing animal studies that contributed to the discovery of early biomarkers of AKI."
Response:
We are grateful for this comment. We have added references to pivotal animal studies (Line 43) to provide the necessary historical context on the discovery and biological plausibility of these biomarkers.

Reviewer:
"The introductory part itself should be expanded and further clarified the importance of incorporating new biomarkers into clinical practice."
Response:
Thank you for pointing out this issue. We expanded the Introduction to explicitly articulate the clinical opportunity missed by traditional markers and the necessity of new biomarkers for timely intervention.

Reviewer:
"I am missing an explanation of the introduction of the panel into clinical practice. The answer to the question 'Why is it important?'"
Response:
Thank you for this valuable comment. We clarified Section 4 to explain that panels are crucial for simultaneously assessing different injury mechanisms, thereby increasing diagnostic precision and guiding tailored therapy.

Reviewer:
"Table 2 is not clear... It is suggested that the 'center' format should not be used, but the Justify format."
Response:
We appreciate you pointing out this issue. We have reformatted and restructured Table 2 to clearly separate columns and rows and improve readability.

Reviewer:
"The text should emphasize the importance of harmonizing and coordinating The KDIGO guidelines with clinical practice."
Response:
We want to thank the Reviewer for this comment. We've strengthened the discussion on harmonizing KDIGO guidelines with clinical practice, citing trials (BigpAK-2) that demonstrate the efficacy of biomarker-guided care bundles.

Reviewer:
"Is there a possibility of a 'cost and benefit' analysis... the data on the cost of introducing these protocols is very important."
Response:
Thank you for providing insightful comments. We've expanded Section 6 to address cost-effectiveness, referencing NICE guidelines and recent economic data from trials to discuss the financial viability of implementation.

We hope that the revisions and detailed responses provided here satisfactorily address the Reviewer's concerns. We thank you again for your time and valuable feedback, which has significantly strengthened our manuscript. We look forward to your positive response.

Round 2

Reviewer 1 Report

Comments and Suggestions for Authors

Many thanks for your modifications on this review paper on established and novel AKI markers. I still believe that you could have mentioned first the classification you are using but then present them in an order from the most to least studied, for a busy clinician that means they can read quickly through the first few. However, I respect your choice to present them as such. 

Author Response

Dear Reviewer,
Thank you for your valuable feedback.

Upon further reflection, we have decided to follow your suggestion and have reordered the biomarkers in the manuscript to prioritize the most established ones, improving readability for busy clinicians. Additionally, we have added a new section regarding CXCL-9, have revised Table 3 to reflect these updates and have added minor corrections in another manuscript's sections.

We appreciate your insights, which have helped improve the quality of this review.